# Comparing BeadChip and WGS Genotyping: Non-Technical Failed Calling Is Attributable to Additional Variation within the Probe Target Sequence

**DOI:** 10.3390/genes13030485

**Published:** 2022-03-09

**Authors:** Moran Gershoni, Andrey Shirak, Rotem Raz, Eyal Seroussi

**Affiliations:** Agricultural Research Organization (ARO), Volcani Center, Institute of Animal Science, HaMaccabim Road, P.O. Box 15159, Rishon LeTsiyon 7528809, Israel; gmoran@volcani.agri.gov.il (M.G.); shiraka@volcani.agri.gov.il (A.S.); rotemv081@gmail.com (R.R.)

**Keywords:** genomic evaluation, genotyping platforms, single nucleotide polymorphism

## Abstract

Microarray-based genomic selection is a central tool to increase the genetic gain of economically significant traits in dairy cattle. Yet, the effectivity of this tool is slightly limited, as estimates based on genotype data only partially explain the observed heritability. In the analysis of the genomes of 17 Israeli Holstein bulls, we compared genotyping accuracy between whole-genome sequencing (WGS) and microarray-based techniques. Using the standard GATK pipeline, the short-variant discovery within sequence reads mapped to the reference genome (ARS-UCD1.2) was compared to the genotypes from Illumina BovineSNP50 BeadChip and to an alternative method, which computationally mimics the hybridization procedure by mapping reads to 50 bp spanning the BeadChip source sequences. The number of mismatches between the BeadChip and WGS genotypes was low (0.2%). However, 17,197 (40% of the informative SNPs) had extra variation within 50 bp of the targeted SNP site, which might interfere with hybridization-based genotyping. Consequently, with respect to genotyping errors, BeadChip varied significantly and systematically from WGS genotyping, introducing null allele-like effects and Mendelian errors (<0.5%), whereas the GATK algorithm of local de novo assembly of haplotypes successfully resolved the genotypes in the extra-variable regions. These findings suggest that the microarray design should avoid polymorphic genomic regions that are prone to extra variation and that WGS data may be used to resolve erroneous genotyping, which may partially explain missing heritability.

## 1. Introduction

On the eve of the genomic era, it was reasoned that progress in the use of DNA technologies for the enhancement of cattle production would be proportional to advances in the knowledge of the underlying mechanisms of the genes involved [1]. However, with a few exceptions of variations in major genes, such as *DGAT* and *ABCG2* [2,3], which have little effect on net merit indexes, it was soon realized that the effects on dairy production traits better fit the infinitesimal model for quantitative traits of Fisher [4]. Giving up on exact identification of affected genes, genomic selection based on SNP information has been successfully used as an “improved black-box approach” to predict the genomic values of these traits [5]. Genomic selection increases the genetic gain of economically significant traits by shortening the generation interval, improving the accurate selection of sires for artificial insemination and enhancing the intensity of their selection [6].

Two commercial companies, Illumina and Affymetrix, have developed SNP genotyping technologies on a large scale. The Illumina microarray uses microscopic beads, whereas Affymetrix uses glass or silicon microchip surfaces. Illumina beads are applied into microarray wells allowing multiple (~30) technical replicates. Depending on the targeted variation, two designs of beads are used. For the first design (D1 type), which commonly targets the variation A/G, the beads contain a 50 bp oligonucleotide for which the 3′ end is positioned directly adjacent to the SNP site. The second design (D2 type) requires two types of beads for which the 3′ end of the 50 bp oligonucleotide overlaps with polymorphic site. This design, which is normally used for A/T or C/G variations, allows polymerase extension only when a perfect match occurs. DNA samples are amplified, digested into smaller parts and hybridized to the beads’ probes. Then, an extension reaction is performed, elongating the samples by a labeled nucleotide [7]. Affymetrix chips are fabricated by the in situ synthesis of 25–70 bp oligonucleotide probes. Following a restriction enzyme digestion, DNA samples are ligated to adapters, amplified, fluorescently labelled, and hybridized to the probes [8]. In both technologies, the chips are eventually evaluated by laser and the sample genotypes are then computed using specialized software.

Aiming at developing a robust genotyping platform for genomic selection of dairy cattle, a U.S.-based consortium has established a panel of markers for the Illumina technology. The resulting BovineSNP50 BeadChip was commercially released in 2007, offering the detection of 54,001 SNPs, and its marker design was based on the alignment of sequence data from different breeds of cattle to a single cow’s genome assembly: the Hereford cow, L1 Dominette [9]. To date, several versions of this most widely used chip have been made available. The current third version offers detection of 53,218 SNPs that are uniformly distributed across the cattle genome. Frequently inseminated with U.S. bull semen, the Israeli Holstein population has been under intensive selection for 60 years and its genetic pool has similar characteristics to the U.S. Holstein. Coping with a hot and humid climate, the centralized Israeli Holstein breeding program introduced genomic selection in 2008, adopting the Illumina BeadChip genotyping technology to evaluate the traits that compose the Israeli selection index, including kg milk, kg fat, kg protein, somatic cell score, daughters’ fertility, herd life, persistency, dystocia, and calf mortality [10]. Since then, several additional BeadChip versions based on the BovineSNP50 have been used, including GeenSeek GGP Bovine 150 K, designed to detect 138,974 SNPs, and Illumina’s BovineLD (7931 SNPs) and BovineHD chips (777,962 SNPs) [11]. With the increasing availability of whole-genome sequencing (WGS), it is now possible to implement genotype calling to larger numbers of SNPs and compare genotyping precision between WGS and BeadChip techniques [12]. In this work, we compare the WGS genotypes of 50,392 genomic positions to their BovineSNP50 counterparts to determine the level of agreement between hybridization-based and sequencing-based genotyping.

## 2. Materials and Methods

### 2.1. DNA Extraction and Sample Preparation for WGS

Sperm DNA from 17 bulls was extracted according to the manufacturer’s protocol of the Genomic Mini AX Swab & Semen Spin kit (#025-100S, A&A Biotechnology, Gdynia, Poland). By direct pedigree analysis, this bull sample was chosen to represent several lineages and, when available, several bulls from the same lineage were included to allow testing for allele segregation. Generally, as indicated by analysis of the kinship-coefficient average and standard deviation, the level of pedigree relationships among these bulls was 0.027 ± 0.04 (Appendix A). Whole-genome sequencing was performed with Illumina HiSeq X Ten (90 Gb), or with NovaSeq platforms, by the sequencing service of TheragenEtex (Suwon, South Korea). It should be noted that the DNA sequencing and BeadChip genotyping of the selected bulls provided optimal results with respect to technical issues, such as DNA quality and handling.

### 2.2. WGS Analysis for the VCF Method

Following Genome Analysis Toolkit (GATK) best practice workflows of the Broad Institute [13], we applied a short-variant discovery within the sequence reads mapped to the ARS-UCD1.2 reference genome [14]. Because the pipeline used produces a Variant Call Format (VCF) text file, which stores all genomic positions with sequence variations, we refer to this procedure as the VCF method. For this method, variant calling was conducted as we previously described [15]. Briefly, adapter sequences and low-quality tails of reads were removed with the software Trimmomatic [16]. Raw reads were then aligned to the reference genome (ARS-UCD1.2 [14]) with BWA-MEM [17]. To avoid biases introduced by data generation steps, such as PCR amplification, PCR duplicates were removed using Picard tools (version 2.20.2; http://broadinstitute.github.io/picard/, accessed on 13 February 2022), and curated BAM files were coordinated, sorted, and indexed by Picard tools. Variant calling and filtration stages were performed with GATK 3.6 [13], via local re-assembly of haplotypes by the GATK HaplotypeCaller algorithm to generate genomic VCF files (GVCFs). Using the GenotypeGVCFs algorithm, the GVCF results from 17 bulls were jointly called to produce a combined GVCF file. Variant calls with mean quality score (MQ) >30 were retained for the downstream comparisons and IGV was used to visualize mapped reads [18].

### 2.3. WGS Analysis for GAP5 Method

Based on direct read aligning to 50 bp sequences spanning the SNP top genomic sequences in the BovineHD manifest file, we also devised a simpler method to directly call the SNP alleles using the GAP5 sequence assembly visualizer. In this procedure, referred to as the GAP5 method, BWA-MEM was used for read mapping to genomic templates by preparing a temp_cons file (fastq format, Appendix A) containing 100,784 contig templates. Each contig consisted of 252 bp in which the central 50 bp were of the targeted allele and the remaining bases were annotated as Ns. Containing the compressed sequence reads, fastq.gz format files were aligned to genomic templates using the following command line: “bwa mem -t24 -k50 -w0 -d0 -r150 -c1000000 -D1 -W50 -m0 -S -P -A1 -B150 -O150 -E150 -L150 -U0 -a -T50 -h1000000 -Y temp_cons sire.R1.fq.gz > sire.R1.sam”. Using the SAMtools software package, the resulting sam-formatted files were converted to bam format (“samtools view -b -S sire.R1.sam > sire.R1.bam”) and sorted (“samtools sort sire.bam sire.R1.srt”). The sorted file was imported into a GAP5 database by merging the contigs to their temporary templates (temp.g5d and temp.g5x, Appendix A). The content of GAP5 contig list, including the R1 read counts, was then copied into an Excel sheet to be merged with the R2 output and to be further analyzed. For each SNP, the threshold for allele detection was set to the number of total hits divided by an empirical coefficient (4.28).

### 2.4. Statistics

BeadChip genotypes were compared to WGS genotypes obtained by the VCF and the GAP5 methods, and the differences between these methods’ mean numbers of concordant and non-concordant genotypes were examined and analyzed for statistical significance using a paired *t*-test for a difference in means as calculated by the Excel T.TEST function with two-tailed distribution and paired test options. To compare the mean coverage between platforms, this function was used with two-tailed distribution and two-sample unequal-variance options. With respect to the genotyping concordance analyses, autosomal and X-chromosomal markers were similarly treated.

## 3. Results

### 3.1. Selection of Polymorphic Sites and Genomic Coordinates

Based on the data records of the Israeli herdbook, we selected 50,392 SNPs that were present in both BovineHD and the early versions of BovineSNP50 microarrays (Appendix A). Most of the selected markers were of the D1 type and only 1870 were of D2. The BovineHD manifest file and its updates provide the SNP genomic coordinates based on the old genome builds (UMD3.1 or Btau4.2), whereas ARS-UCD1.2 is the current reference genome. To obtain the ARS-UCD1.2 coordinates (Appendix A), we batch-BLASTN searched this build querying each SNP by its top genomic sequence described in this manifest file. The obtained positions matched those previously published [19], except for 308 SNPs (Appendix A).

### 3.2. WGS and Variant Calling from Next-Generation Sequencing (NGS) Data

Using a 151 bp paired-end setup, we deep-sequenced 6 and 11 genomes using the NovaSeq 6000 and HiSeq X platforms, respectively. With a mean coverage depth of ~30-fold, the HiSeq platform produced significantly less (~25%) sequence reads than NovaSeq, which yielded a mean coverage depth of ~40-fold (Table 1). Following GATK best practice workflows of the Broad Institute [13], we applied a short-variant discovery within the sequence reads mapped to the ARS-UCD1.2 reference genome. For the 17 sires, we compared the BeadChip genotypes of the Israeli herdbook to those called by the VCF and GAP5 methods (Table 1). The NovaSeq system provided a significantly (*p* = 9.8 × 10^−5^) larger fold coverage (~35%). However, as we used this method to genotype older sires, the mean rate of matches to genotypes from the older BeadChip was reduced. To exclude the possibility of sequencing method bias, sire 7733 was sequenced in both platforms, which yielded a similar number of matches indicating no critical bias. Thus, comparing to HiSeq, the number of NovaSeq mismatches for both calling methods was also reduced (Table 1), indicating that the problem was related to BeadChip representation bias rather than a lower sequence quality. For both technologies, the number of mismatches between BeadChip and WGS genotype was low (0.2%, Table 1). However, although the VCF method is known to provide more accurate variant calling [20], for the HiSeq results, it pointed to ~8-fold more mismatches than the GAP5 output.

### 3.3. Case Analyses of Beadchip and WGS Genotype Mismatches

To determine why certain SNP yielded discordant BeadChip and WGS genotypes, we used assembly visualizers to examine their sequences at the read level. As shown in Figure 1, the allele that was undetectable by BeadChip and GAP5 genotyping carried an additional variation close to the SNP site (T variant 3 bp upstream of the G allele, Figure 1a). Such additional variation within the probe target sequence is likely to interfere with the hybridization to the Illumina bead. Similarly, this extra variation prevents the mapping of reads to the GAP5 contigs, which requires a full match with a minimum score 50 to output (Figure 1b). To further assess the possible effect of additional variation within the probe target sequence, we examined our VCF files for such variations within 50 bp upstream and downstream of the SNP positions (Table 2). Table 2 reports as many as 42,848 polymorphic SNPs within the 17 bull sample, whereas the numbers presented for the HS samples were higher (up to 44,890, Table 1), as the latter include the analysis of the concordant genotyping also for the SNPs that were not polymorphic within the examined sample. For the polymorphic SNPs, results showed that about 1% of the BeadChip sites were not detected as SNPs in at least one of the genomes. That is because the alleles recorded in those genomic positions were not only the two alternate bases annotated in the BovineHD manifest file (Table 2). Of the BeadChip SNPs that displayed an extra variation, it was estimated that the genotyping of as much as ~25% might be affected by this problem in one or more of the sires, as miscalling may arise from variation in only one of the alternative directions (Table 2).

To further examine the effect of a third allele on BeadChip genotyping, we examined cases that did not fit the Mendelian paradigm, as the father allele of a homozygotic genotype was not detected in the son. As shown in Figure 1c, for the marker BTB-01793064, the BeadChip genotypes were GG for the father and AA for the son, whereas the VCF complementary genotypes were AG and AA, respectively. However, the A allele detected by the VCF method was a third allele, as it had a 5 bp deletion within the targeted 50 bp and, therefore, it differed from the A allele described in the BeadChip setup file. In homozygous state, the BeadChip hybridization recognized this mutated A allele in the son; however, in the heterozygous state, the hybridization levels of this deletion were reduced and it was not called in the father. For the described father–son pair, we detected 255 SNPs (0.5%) for which BeadChip genotypes did not follow the Mendelian rules. These results suggest that the development of BeadChip markers in the polymorphic genomic regions that are prone to extra variation within the probe target introduces systematic genotyping inaccuracies that negatively affect the statistical power of the data obtained from SNP microarrays.

## 4. Discussion

The availability of WGS allows the comparison of genotyping accuracy between WGS and BeadChip techniques. In this paper, we present the deep sequencing of 17 genomes of Israeli Holstein sires belonging to the bull panel used for the artificial insemination of dairy cows. In both the HiSeq and NovaSeq platforms, the 151 bp paired-end reads provided detailed information that allowed accurate alignment to the reference genome. Based on this alignment and following the best practices provided by the GATK software, SNPs and indels were simultaneously called via the local de novo assembly of haplotypes in the variable regions and the genotypes were recorded in VCF files [13]. Providing that the reference genome is compatible with that of the called individual, this VCF method yields the most accurate description of the genomic variation. However, the current reference genome (ARS-UCD1.2) is based on beef cattle, which somewhat differs from the Holstein genome. For example, the genomic sequence described in the BeadChip manifest file for ARS-BFGL-NGS-2370 does not match anywhere within the current genome build (Appendix A). On the other hand, BeadChip calling is based on a probe sequence of only 50 bp and a hybridization process that is sensitive to unexpected nucleotide variation within the probe target sequence. To somewhat mimic this hybridization procedure computationally, we designed a calling method in which the reads were mapped to 50 bp spanning the SNP site using the source genomic sequence that is available in the BovineHD manifest file (Appendix A). Ideally, this method should have relied on the probe sequences, in which the called allele is encoded at the very 3′ end. However, this proved to be tricky, as current bioinformatic tools, such as BWA [17], were designed to map short reads onto long templates, whereas our method required mapping reads that were three times longer than the 50 bp target templates. Thus, positioning the SNP in the center of the template was essential to the detection of perfect matches. Moreover, this does not simulate hybridization accurately, as the 3′ end of the probe sequence is far more important than the 5′ end, and as the probe sequence content (i.e., GC%) is of paramount importance in hybridization [21]. Nevertheless, when compared to the VCF method, this method significantly (*p* = 2.9 × 10^−4^) yielded 8% more matching genotypes and significantly (*p* = 2.1 × 10^−7^) 8-fold less mismatches with the BeadChip genotypes (Table 1). Thus, although the GAP5 method accounts for none of the above-mentioned constraints, it output a better simulated BeadChip genotyping. This result arose from the systematic genotyping bias that was introduced by the development of BeadChip markers in the polymorphic genomic regions that are prone to interference by extra variation within the probe target. We showed that in heterozygous state, an allele that does not perfectly match the probe target may behave as a null allele, although it can be readily detected in homozygotes.

Null alleles can generate significant genotyping inaccuracies that negatively affect the statistical power of genetic studies [22,23]. In our relatively small bull sample, as much as 40% of the SNPs had extra variation within 50 bp of the targeted SNP, including variation occurrences at the site, upstream or downstream of it. Thus, the interfering variant may explain the frequent (0.5%) cases of Mendelian errors in BeadChip data. This observation is in line with a previous report of pruning 0.5% of BovineHD SNPs because of their high Mendelian error rate (>0.05) in Holstein cattle [24]. It is important to note that the best practices for processing SNP data recommend the exclusion of markers that are likely to be not correctly called (e.g., showing deviation from the Hardy–Weinberg equilibrium).

Genotyping inaccuracies may also account for some of the missing heritability encountered in SNP-array based studies [25,26]. To call an SNP genotype, the BeadChip technique makes use only of one of the two possible orientations. Thus, in our sample, the extra variation might amount to ~25% of the informative SNPs, arising of unexpected variation at the SNP site (~1%) and interfering flanking variants in one of the orientations (~24%; Table 1). This indicates that state-of-the-art WGS genotyping reflects better the actual genotypes than hybridization-based genotyping. Indeed, considering the missing heritability issue, it was suggested early on that, when the price of sequencing falls, it would be sensible to stop using SNPs and start sequencing whole genomes [25]. Alternatively, it has been shown that, in humans, the imputation of the genotype data can increase the accounted heritability for height and body mass to negligible missing levels [27]. However, current imputation algorithms do not resolve genotyping errors that are produced by interfering sequence variants in hyper-polymorphic regions. It was also shown that exome sequencing provides a better option to the array-based methods [28]. As exons display less divergence in sequence, exome SNPs are likely to reduce extra variation at the probe targets. However, this option is currently commercially available only for human genetics, i.e., Illumina HumanExome BeadChip. Moreover, exome sequencing has several major flaws that may incorporate more ascertainment bias in the discovery of polymorphic markers in cattle. These include allele imbalances and a probe distribution imbalance. Furthermore, a large part of the heritability of complex traits in cattle seems to be due to variation in regulatory non-coding regions [29,30]. Yet, it is noteworthy that most breeding programs nowadays use customized versions of the microarrays that have been adopted for their needs [30] and these carefully designed microarrays may have better targeted causal variants avoiding a hyper-variable genomic hotspot.

## 5. Conclusions

We developed a method to map short sequence reads to 50 bp genomic contigs. The alignment of short-sequence reads to shorter contig templates somewhat mimics hybridization. At SNP sites, the nearby extra variation hinders hybridization-based allele calling and adjacent extra variation is observable in over 40% of SNPs in BovineSNP50 BeadChip. This variation produces interfering sequence variants that induce null allele-like effects and Mendelian errors, which are a primary reason for failed calling that is not related to technical problems, such as DNA quality and handling.

## Figures and Tables

**Figure 1 genes-13-00485-f001:**
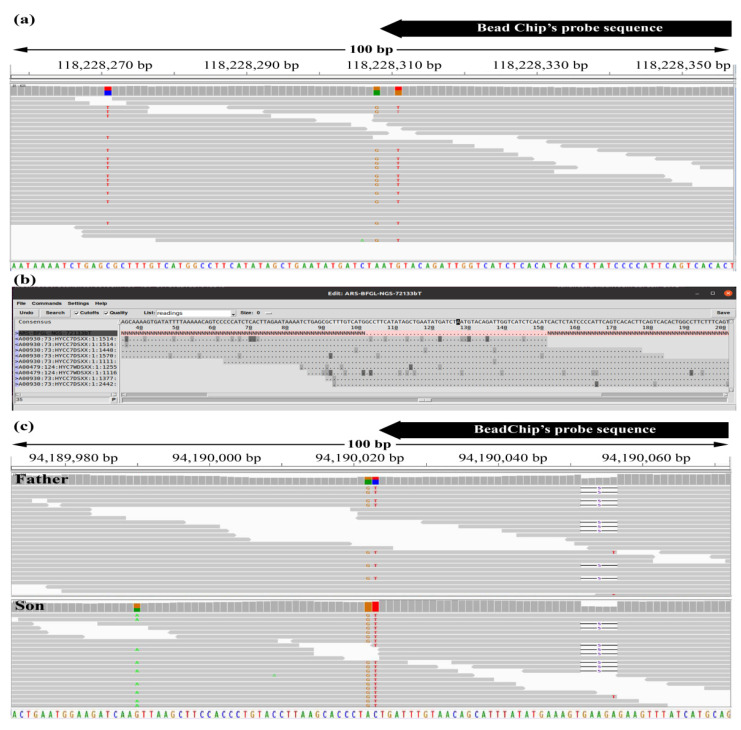
Output of the assembly visualizers for SNPs ARS-BFGL-NGS-72133 and BTB-01793064. Position 118,228,308 on BTA1 of sire 7424 genome was examined. (**a**) An IGV output. Using the VCF method, this SNP was genotyped as heterozygous with 17 and 13 reads of A and of G alleles, respectively, whereas the BeadChip and GAP5 genotypes were AA homozygous. (**b**) GAP5 output with 9 reads of the A allele. Pink background denotes the template with 50 bp spanning the SNP site (black background on the consensus line). Bases identical to the consensus sequence are denoted as dots with background color that corresponds to the quality score, with light gray indicating higher quality. (**c**) Position 94,190,023 on BTA1 of father and son (7396 and 7851, respectively) genomes were examined. Using the VCF method, the father’s SNP was genotyped as heterozygous with 12 and 7 reads of C and of T alleles, respectively, whereas the son was homozygous for the T allele (19 reads).

**Table 1 genes-13-00485-t001:** Comparison between BeadChip and WGS genotyping.

Sire	NGS ^1^	# of Spots ^2^	Match GAP5 ^3^	Match VCF ^3^	Miss GAP5 ^4^	Miss VCF ^4^
3376	NS	401,967,974	34,493	35,030	3	25
3651	NS	361,519,877	34,452	35,081	3	13
3756	NS	449,885,842	34,507	35,033	6	18
3811	NS	399,980,764	34,543	35,061	0	17
7165	NS	382,156,099	34,397	35,055	5	21
7592	NS	458,256,777	33,397	33,796	0	9
7733	NS	439,943,852	42,542	38,845	16	81
**Mean ± SE ^5^**		**413,387,312 ± 13,824,080**	**35,476 ± 1188**	**35,415 ± 599**	**4.7 ± 2.1**	**26.3 ± 9.3**
7396	HS	310,285,759	44,238	40,753	15	54
7400	HS	319,575,305	43,364	41,313	5	67
7424	HS	337,946,649	44,879	41,136	5	57
7510	HS	300,382,216	44,890	41,376	2	62
7559	HS	273,572,927	44,626	41,221	6	62
7679	HS	285,115,013	43,762	40,470	8	62
7733	HS	302,259,900	42,224	38,849	16	78
7738	HS	296,384,014	44,778	41,334	6	76
7851	HS	327,126,812	44,945	41,320	5	56
7936	HS	317,888,134	44,721	41,154	7	56
9078	HS	304,472,448	44,890	41,355	10	64
**Mean ± SE ^5^**		**306,819,016 ± 5,585,356**	**44,302 ± 260**	**40,934 ± 225**	**7.7 ± 1.3**	**63.1 ± 2.4**

^1^ WGS was performed on the NovaSeq (NS) and HiSeq (HS) platforms. ^2^ Each spot produced two reads (forward and reverse). ^3^ Match GAP5 and Match VCF represent the number of concordant genotypes between the BeadChip data and the GAP5 and VCF methods, respectively. ^4^ Miss GAP5 and Miss VCF represent the number of non-concordant genotypes between the BeadChip data and the GAP5 and VCF methods, respectively. ^5^ Means and their standard errors are given for each of the platforms (boldface).

**Table 2 genes-13-00485-t002:** BeadChip SNPs with additional variation within the probe target sequence ^1^.

	Total	Polymorphic	Variation Upstream	Variation Downstream	Any Extra Variation	Not SNP ^2^	Potential Problem
# of SNPs	50,392	42,848	10,381	10,237	17,197	393	17,265
%	100	85	24.2	23.9	40.1	0.9	40.3

^1^ Additional variation within 50 bp of an SNP site that was found in the 17 analyzed genomes. ^2^ At the SNP site, more than two alleles, indel, or none single nucleotide variation were found.

## Data Availability

Restrictions apply to the availability of genotyping data. Data were obtained from the Israel Cattle Breeding Association (ICBA) and are available from the authors with the permission of ICBA.

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
