# Peer review of "Comparing BeadChip and WGS Genotyping: Non-Technical Failed Calling Is Attributable to Additional Variation within the Probe Target Sequence"

_genes, 2022, doi:10.3390/genes13030485_

Round 1

Reviewer 1 Report

The manuscript “Comparing BeadChip and WGS Genotyping: Non-technical Failed Calling Is Attributable to Additional Variation within the Probe Target Sequence” describes an effort to document the reason for inconsistent or failed genotype calls on Beadchip genotyping platforms in Israel Holstein populations.  The manuscript is in general well written, and the approaches and interpretations are solid.  I have a few minor suggestions for the authors to consider that may improve clarity of the presented information.

Line(s)                  Comment

23                           Or perhaps design multiple probes to account for the new variation?

80-81                     Could you provide some indication of the level of pedigree relationships among the 17 chosen bulls?

161 Table 1         Apply the 3 and 4 superscripts to the VCF columns also.

171                         Additional variation not accounted for in the sequence for probe design?

216                         and/or requires careful editing for those SNP impacted.

                                This is reminiscent of approaches used to detect SNP under expression probe sequences using Affymetrix probe designs and divergent breeds or populations that differ from the reference sequence (for example BMC Genomics BMC Genomics  9:252.  2008).

225-227                This is a good argument for Pan-genome assemblies to be generated.

241-242                “… do not simulated hybridization …” ?? Re-write as this sentence does not make sense as it is written.

256                         variation

262                         ~25% of what?

Author Response

The manuscript “Comparing BeadChip and WGS Genotyping: Non-technical Failed Calling Is Attributable to Additional Variation within the Probe Target Sequence” describes an effort to document the reason for inconsistent or failed genotype calls on Beadchip genotyping platforms in Israel Holstein populations. The manuscript is in general well written, and the approaches and interpretations are solid. I have a few minor suggestions for the authors to consider that may improve clarity of the presented information.

Line(s) Comment

23 Or perhaps design multiple probes to account for the new variation?
Designing multiple probes that account for all variation would be only possible if the population would be well characterized on the nucleotide level.
At this stage, we feel that our recommendation to altogether avoid repetitive regions is better.

80-81 Could you provide some indication of the level of pedigree relationships among the 17 chosen bulls?
In the appendix we added a detailed analysis of the kinship among the 17 chosen bulls. We now refer to this analysis in the text:
"Generally, as indicated by analysis of the kinship-coefficient average and standard deviation, the level of pedigree relationships among these bulls was 0.027±0.04 (Figure S1)."

161 Table 1 Apply the 3 and 4 superscripts to the VCF columns also.
Done.

171 Additional variation not accounted for in the sequence for probe design?
The reviewer refer to the title of the section "Case analyses of BeadChip and WGS genotype mismatches" suggesting to name it after the conclusion derived. We think that such revision would suggest that we were biased towards this hypothesis, whereas, our analysis consisted on unbiased approach. Using the genome viewer, we simply observed what were the specific sequences involved in multiple cases that lead to Mendelian inconsistencies and then we concluded that these generally aroused from additional variation that was not accounted for in the sequence for probe design.

216 and/or requires careful editing for those SNP impacted.
This is reminiscent of approaches used to detect SNP under expression probe sequences using Affymetrix probe designs and divergent breeds or populations that differ from the reference sequence (for example BMC Genomics BMC Genomics 9:252. 2008).
We assume that the referee suggests adding a conclusion that the SNPs impacted by interfering extra variation should be excluded from analyses. However, this better fits Discussion and not Results. We added in Discussion: "It is important to note that best practices for processing SNP data recommend exclusion of markers that are likely to be not correctly called (e.g., showing
deviation from Hardy-Weinberg equilibrium)."

225-227 This is a good argument for Pan-genome assemblies to be generated.
We agree. Yet, it is not in the focus of our manuscript to promote this.

241-242 “… do not simulated hybridization …” ?? Re-write as this sentence does not make sense as it is written.
Indeed, the subject was missing from this sentence. We rephrased:
"Moreover, this does not simulate hybridization accurately as the 3’ end of the probe sequence is far more important than the 5’ end, and as the probe sequence content (i.e., GC%) is of paramount importance in hybridization [21]."

256 variation
We added: "including variation occurrences at the site, upstream or downstream of it."

262 ~25% of what?
We added: "extra variation might amount to ~25% of the informative SNPs,

Reviewer 2 Report

The research article “Comparing BeadChip and WGS Genotyping: Non-technical failed calling is attributable to additional variation within the probe target sequence” aims to highlight an area of technical limitation in microarray-based genomic selection in dairy cattle, that could potentially explain observed missing heritability. The authors use whole-genome sequences of 17 cattle with different alignment approaches to compare, contrast and highlight the issue of extra variation in the template DNA that could potentially interfere with genotyping by probe hybridization in BeadChip data. The research is well designed and carried out. 

Major Revisions

None suggested

Minor Revisions

  • Page number: 7, lines 241-243: The sentence reading “Moreover, do not simulated hybridization …..” may have typos or something missing and requires correction.

Author Response

The research article “Comparing BeadChip and WGS Genotyping: Non-technical failed calling is attributable to additional variation within the probe target sequence” aims to highlight an area of technical limitation in microarray-based genomic selection in dairy cattle, that could potentially explain observed missing heritability. The authors use whole-genome sequences of 17 cattle with different alignment approaches to compare, contrast and highlight the issue of extra variation in the template DNA that could potentially interfere with genotyping by probe hybridization in BeadChip data. The research is well designed and carried out.

Major Revisions- None suggested

Minor Revisions
Page number: 7, lines 241-243: The sentence reading “Moreover, do not simulated hybridization …..” may have typos or something missing and requires correction.
Indeed, the subject was missing from this sentence. We rephrased: "Moreover, this does not simulate hybridization accurately as the 3’ end of the probe sequence is far more important than the 5’ end, and as the probe sequence content (i.e., GC%) is of paramount importance in hybridization [21]."

Reviewer 3 Report

The authors sequenced 17 Israeli Holstein bulls using WGS and microarray Beadchip, and found that, although the agreement between microarray and WGS calls are high for those that were sequenced in the microarray, WGS have revealed 40% of extra variation within 50 bp of the microarray target sites. The authors propose that this difference could result in the missing heritability estimated from genotype data alone. To perform the analysis, a method was described to map short sequence reads from WGS (of ~150 bp) to 50 bp genomic contigs from microarray sequencing. Overall the methods and results are clearly presented, and the conclusion  supported.

However, I find it unsurprising that the genotype data alone does not explain missing heritability in bulls, as this has already been known in humans. It has also been shown that imputation of the genotype data can increase the accounted heritability for some human traits to negligible missing level (Yant et.al. https://doi.org/10.1038/ng.3390). I think commenting on whether imputation can or cannot account for heritability for bulls (as compared to human) would make the paper more valuable. Further, it would be useful to comment on which traits the genotyping array used (BovineSNP50) was intended to cover by design.

Author Response

The authors sequenced 17 Israeli Holstein bulls using WGS and microarray Beadchip, and found that, although the agreement between microarray and WGS calls are high for those that were sequenced in the microarray, WGS have revealed 40% of extra variation within 50 bp of the microarray target sites. The authors propose that this difference could result in the missing heritability estimated from genotype data alone. To perform the analysis, a method was described to map short sequence reads from WGS (of ~150 bp) to 50 bp genomic contigs from microarray sequencing. Overall the methods and results are clearly presented, and the conclusion supported.
However, I find it unsurprising that the genotype data alone does not explain missing heritability in bulls, as this has already been known in humans. It has also been shown that imputation of the genotype data can increase the accounted heritability for some human traits to negligible missing level (Yant et.al. https://doi.org/10.1038/ng.3390). I think commenting on whether imputation can or cannot account for heritability for bulls (as compared to human) would make the paper more valuable.
We added and rephrased: "Alternatively, it has been shown that in humans, imputation of the genotype data can increase the accounted heritability for height and body mass to negligible missing levels [27]. However, current imputation algorithms do not resolve genotyping errors that are produced by interfering sequence variants in hyper-polymorphic regions."

Further, it would be useful to comment on which traits the genotyping array used (BovineSNP50) was intended to cover by design.
We added in introduction: "to evaluate the traits that compose the Israeli selection index including kg milk, kg fat, kg protein, somatic cell score, daughters’ fertility, herd life, persistency, dystocia, and calf mortality".
